# The Effects of the Pneumonia Lung Microenvironment on MSC Function

**DOI:** 10.3390/cells13181581

**Published:** 2024-09-20

**Authors:** Lanzhi Liu, Juan Fandiño, Sean D. McCarthy, Claire H. Masterson, Ignacio Sallent, Shanshan Du, Abigail Warren, John G. Laffey, Daniel O’Toole

**Affiliations:** 1CÚRAM Institute for Medical Devices, University of Galway, H91 W2TY Galway, Ireland; l.liu8@universityofgalway.ie (L.L.); juan.fandinogomez@universityofgalway.ie (J.F.); sean.mccarthy@universityofgalway.ie (S.D.M.); claire.masterson@universityofgalway.ie (C.H.M.); ignacio.sallent@universityofgalway.ie (I.S.); shanshan.du@universityofgalway.ie (S.D.); a.warren4@universityofgalway.ie (A.W.); john.laffey@universityofgalway.ie (J.G.L.); 2Discipline of Physiology, University of Galway, H91 W5P7 Galway, Ireland; 3School of Medicine, University of Galway, H91 W5P7 Galway, Ireland; 4Discipline of Anaesthesia, University of Galway, H91 V4AY Galway, Ireland; 5Anaesthesia and Critical Care, Galway University Hospital, H91 V4AY Galway, Ireland

**Keywords:** mesenchymal stem cell (MSC), lung microenvironment, inflammation

## Abstract

Background: Despite promise in preclinical models of acute respiratory distress syndrome (ARDS), mesenchymal stem cells (MSC) have failed to translate to therapeutic benefit in clinical trials. The MSC is a live cell medicine and interacts with the patient’s disease state. Here, we explored this interaction, seeking to devise strategies to enhance MSC therapeutic function. Methods: Human bone-marrow-derived MSCs were exposed to lung homogenate from healthy and *E. coli*-induced ARDS rat models. Apoptosis and functional assays of the MSCs were performed. Results: The ARDS model showed reduced arterial oxygenation, decreased lung compliance and an inflammatory microenvironment compared to controls. MSCs underwent more apoptosis after stimulation by lung homogenate from controls compared to *E. coli*, which may explain why MSCs persist longer in ARDS subjects after administration. Changes in expression of cell surface markers and cytokines were associated with lung homogenate from different groups. The anti-microbial effects of MSCs did not change with the stimulation. Moreover, the conditioned media from lung-homogenate-stimulated MSCs inhibited T-cell proliferation. Conclusions: These findings suggest that the ARDS microenvironment plays an important role in the MSC’s therapeutic mechanism of action, and changes can inform strategies to modulate MSC-based cell therapy for ARDS.

## 1. Introduction

Acute respiratory distress syndrome (ARDS) is a severe disease with high mortality, characterized by the presence of pulmonary oedema, hypoxemia, and mechanical ventilation. ARDS is triggered by pneumonia, sepsis, gastric aspiration or severe trauma [1]. Different mechanisms contribute to the pathophysiology of ARDS, including endothelial permeability, alveolar epithelial injury and dysfunction, lung inflammation and mechanical stress [2]. Mesenchymal stromal cells (MSCs) are being considered as a novel therapy for ARDS, showing efficacy in alleviating severity and improving survival in various ARDS models [3]. The mechanisms involved have been widely investigated, including improving alveolar epithelium function [4], reducing lung bacterial burden [5], inhibiting inflammation by modulating macrophages and T cells function [6,7], and reversing epithelial-mesenchymal transition [8]. However, while the results from clinical trials have shown the safety of MSC-based therapy in ARDS, there is no demonstration of efficacy [9,10,11]. To improve the outcomes in clinical studies, the fate and function of the infused MSCs in the lung microenvironment need to be investigated.

Growing evidence has shown that the host microenvironment modulates MSC viability, phenotype and secretome profile, especially in an inflammatory milieu [12]. After exposure to pooled serum containing high levels of IL-8, IL-6, and IL-1RA from ARDS subjects, MSCs presented an augmented therapeutic effect with an increase in the secretion of anti-inflammatory mediators such as IL-10 and IL-1RA [13,14]. Besides serum, ex vivo exposure to pooled or individual bronchoalveolar lavage fluid (BALF) from healthy controls or ARDS patients or other non-ARDS lung disease patients also differentially alters the behaviour of MSCs and activates MSCs to produce a spectrum of cytokines [6,15,16]. However, another study found that the lung microenvironment not only enhances some but can also reduce other MSC therapeutic effects. The researchers found that different lung injury models had distinct proteomic profiles with different levels of IL-6, which may lead to different responses to MSC cell therapy, and these detrimental effects can be restored by correcting the microenvironment with glutathione peroxidase-1 or genetically editing MSCs to carry IL-10 or hepatocyte growth factor gene expression [17]. Plasma from ARDS patients can also increase the expressions of IL-6 and N-cadherin in co-cultures of MSCs and lung epithelial cells [17]. Moreover, MSCs undergo apoptosis after being injected, and are phagocytosed by monocytes/macrophages to exert anti-inflammatory effects, leading to IL-10 induction and enhanced phagocytosis by macrophages, contributing to inflammation resolution and tissue repair [18,19,20]. Overall, these studies suggest that lung inflammatory factors, like IL-6 and macrophages, and dysfunction of lung epithelial cells can alter MSC activity, indicating that the identification of disease microenvironments and the modification of naïve MSCs are both crucial in the development of effective MSC therapy for ARDS. However, there is still a gap of knowledge in understanding the crosstalk between infused MSCs and the lung microenvironment, so new in vitro and in vivo models are necessary.

In this study, we aimed to determine the effects of a new aspect of normal and ARDS lung microenvironments on an administered MSC therapeutic. These findings could explain the mechanism of action of the MSC in preclinical models, why the microenvironment is important in their function, and inform future enhancement strategies to finally deliver on the MSC’s promise for ARDS patients.

## 2. Materials and Methods

### 2.1. Cells

Human bone-marrow-derived MSCs (BM-MSCs) were isolated from healthy donors with informed consent under the approval of the Galway University Hospital Clinical Research Ethics Committee (Ethics Ref. C.A.02/08), as previously described and characterized by having positive expression of CD73, CD105, CD90, and MHCI and negative expression of CD34, CD45, CD80, and CD86 [21]. BM-MSCs were cultured to 80% confluence and used in passage 3.

### 2.2. Animal Model Establishment and Preparation of Lung Homogenate

#### 2.2.1. Sham and *E. coli*-Induced Lung Injury Model

All animal work was approved by the Animal Care Research Ethics Committee of the University of Galway and conducted under license from the Health Products Regulatory Authority, Ireland (AE19125/P111). Specific-pathogen-free adult male Sprague Dawley rats (9–13 weeks old, Envigo, Kent, UK) weighing between 350 g and 450 g were used in animal experiments. *Escherichia coli* (*E. coli*)-induced lung injury was initiated as previously described [5]. Briefly, animals were anaesthetized with inhalational isoflurane and 300 μL Dulbecco′s phosphate buffered saline (DPBS) (Thermo Fisher, Dublin, Ireland) with or without 1 × 10^9^
*E. coli* (NCTC, E5162, serotype: O9 K30 H10) was intratracheally instilled via a 16G cannula (sham group, *n* = 11, *E. coli* group, *n* = 13).

#### 2.2.2. Lung Injury Assessment and Ex Vivo Analyses

Forty-eight hours after instillation, lung injury assessment and ex vivo analyses were performed as previously described [5]. Briefly, static lung compliance and arterial blood gas analysis were measured under anaesthesia and after exsanguination, the heart–lung block was dissected from the thorax. One-third of the right caudal lobe was weighed and used to make lung homogenate, after which bronchial alveolar lavage (BAL) was performed on the remaining lung. The BAL fluid was then centrifuged, and the supernatant was stored at −80 °C for protein quantification with Pierce™ BCA Protein Assay Kit (# 23227, Thermo Fisher, Dublin, Ireland), and cytokine-induced neutrophil chemoattractant 1 (CINC-1, cat. no. DY515 R&D Systems™, Abingdon, UK), matrix metalloproteinase 9 (MMP-9, cat. no. DY8174-05 R&D Systems™, Abingdon, UK), and interleukin 6 (IL-6, cat. no. DY506 R&D Systems™, Abingdon, UK) were measured by ELISA.

#### 2.2.3. Lung Homogenate Preparation

Lung homogenate was prepared to mimic the lung microenvironment ex vivo. The weighed tissue was minced, snap-frozen, and stored at −80 °C. DPBS was added at 1 mL per 100 mg of lung tissue, followed by complete homogenization using an Ultra-Turrax disperser (IKA Ltd., Oxford, UK). The homogenate was centrifuged at 500× *g* for 10 min and strained through a 40 µm cell filter. Lung homogenates of 2–4 random animals from the sham group (*n* = 11) or *E. coli* group (*n* = 13) were pooled together to make three pairs, aliquoted and preserved at −80 °C. Protein concentration of the lung homogenates pools was measured with Pierce™ BCA Protein Assay Kit (cat. no. 23227, Thermo Fisher, Dublin, Ireland).

### 2.3. MSC Exposure to Lung Homogenate

BM-MSCs were seeded in 6-well plates (Sarstedt Ltd., Wexford, Ireland) at a density of 2 × 10^5^ cells per well and cultured in complete media overnight [MEM-α, GlutaMAX™ media (Gibco # 32561037; Thermo Fisher, Dublin, Ireland) plus 10% foetal bovine serum (FBS) (Sigma-Aldrich, Arklow, Ireland), 1% Penicillin-Streptomycin (Sigma-Aldrich, Arklow, Ireland), and 2.5 ng/mL of fibroblast growth factor 2 (FGF-2) (cat. no. 11343625, Immuno Tools GmbH, Friesoythe, Germany)]. The cell monolayer was washed once with DPBS, and 1 mL of complete media with lung homogenate containing different concentrations of proteins (50 µg/mL, 100 µg/mL, 200 µg/mL) from sham or *E. coli*-induced lung injury groups was then added to the wells. Twenty-four hours after exposure to the lung homogenate, BM-MSCs were trypsinized and collected for flow cytometric analysis.

### 2.4. MSC Apoptosis and Activation after Exposure to Lung Homogenate

Annexin V-APC (cat. no. 640941, BioLegend, Inc., San Diego, CA, USA) and SYTOX™ Green (cat. no. L34951 B, LIVE/DEAD™ Cell Viability Assay Kit; Thermo Fisher, Dublin, Ireland) were used to detect the early or late apoptosis and necrosis of BM-MSCs as per the manufacturer’s instructions. Briefly, the BM-MSCs were stained with Annexin V-APC and Sytox Green in 1X Annexin V binding buffer (10X concentrate composed of a 0.2 µm sterile filtered 0.1 M Hepes (pH 7.4), 1.4 M NaCl, and 25 mM CaCl_2_ solution) for 15 min at room temperature in the dark. Quantitative flow cytometry was performed using a Cytek Northern Lights 3000 (Cytek Biosciences, Inc., Fremont, CA, USA). Data were analysed by Flowjo 10.9.0 (BD Biosciences, Wokingham, UK).

CD54-APC, CD274-PE-Vio^®^ 615, CD119-FITC (cat. no. 130-121-342, 130-122-811, and 130-099-929, respectively; Miltenyi Biotec, Bisley, UK), CD120b-PE, CD200-PE-CY7, and DRAQ7™ (cat. no. 358403, 399805 and 424001, respectively, BioLegend, Inc., San Diego, CA, USA) were used to test the activation of BM-MSCs according to the manufacturers’ instructions. Briefly, the harvested BM-MSCs were stained with all the antibodies in the FACS buffer (DPBS with 5% FBS) for 30 min on ice in the dark. After being washed with FACS buffer once, the BM-MSCs were stained with DRAQ7™ in FACS buffer for 10 min at room temperature in the dark to exclude dead cells. Quantitative flow cytometry was performed using the Cytek Northern Lights 3000. Data were analysed by Flowjo software v10.6.0 (Becton Dickinson & Company, Franklin Lakes, NJ, USA). All antibodies were titrated prior to use.

### 2.5. Preparation and Functional Assays of Conditioned Media

BM-MSCs were seeded in T175 flasks and expanded to 70% confluence, followed by adding different concentrations of lung homogenate protein as previously described. Twenty-four hours after exposure, the lung homogenate was aspirated, cells were washed with DPBS, 15 mL of MEM-α, GlutaMAX™ basal media without FBS, P/S, or FGF-2 was added, and the BM-MSCs were cultured for another 24 h. The media (conditioned media) were collected, aliquoted, and stored at −80 °C for functional assays.

#### 2.5.1. Cytokine Quantification

The MSC conditioned media (MSC-CM) was analysed to quantify different cytokines by ELISA kits, including interleukin 8 (IL-8, cat. no. DY208 R&D Systems™, Abingdon, UK), interleukin 6 (IL-6, cat. no. DY206 R&D Systems™, Abingdon, UK), monocyte chemoattractant protein 1 (MCP-1, cat. no. DY279 R&D Systems™, Abingdon, UK), vascular endothelial growth factor (VEGF-A, cat. no. DY293B R&D Systems™, Abingdon, UK), tumour necrosis factor receptor 1 (TNFR1, cat. no. DY225 R&D Systems™, Abingdon, UK), and transforming growth factor beta-1 (TGF-β1, cat. no. DY240 R&D Systems™, Abingdon, UK).

#### 2.5.2. Anti-Microbial Assays

To assess the anti-microbial properties of the MSC-CM, a previously established assay protocol [22] was followed. In brief, bacterial cryo-beads coated with *E. coli*, *Staphylococcus aureus* (*S. aureus*), and *Klebsiella pneumoniae* (*K. pneumoniae*) were cultured in 3 mL bijouxs of Tryptone Soya Broth (TSB) at 37 °C in an orbital shaker set to 180 RPM for 16 h. The bacteria were pelleted at 2000× *g* for 30 min and resuspended in 1 mL of DPBS. CFUs/µL were calculated by optical density (OD) measurements at 590 nm and compared to a previously established standard curve. Aliquots of 2 × 10^6^ CFU of each bacterium were plated in V-bottomed 96-well plates in a final volume of 200 µL of MSC-CM or MEM-α control. Following 4 h incubation, the bacteria were pelleted and resuspended in 200 µL of DPBS and transferred to a flat-bottomed 96-well plate and OD readings at 590 nm were measured.

#### 2.5.3. T-Cell Proliferation Assay

Peripheral blood mononuclear cells (PBMC) were isolated from buffy coat under the approval of the Irish Blood Transfusion Service (IBTS), Ireland (Approcal #0000RES585). Fifteen mL of histopaque^®^-1077 (cat. no. 10771, Sigma-Aldrich, Arklow, Ireland) was layered over 15 mL of buffy coat, which was previously diluted with 15 mL of DPBS. Samples were centrifuged at 700× *g* for 30 min at 20 °C with the brake off. The layer of mononuclear cells was collected, washed twice with 20 mL Hanks’ Balanced Salt Solution (HBSS) (cat. no. 55021C, Sigma-Aldrich, Arklow, Ireland), and centrifuged at 500× *g* for 10 min at 25 °C. Live/dead ratios were assessed with Trypan Blue exclusion dye.

The PBMCs were stained with CellTrace™ CFSE Cell Proliferation Kit (cat. no. C34554, Thermo Fisher, Dublin, Ireland) as per the manufacturer’s instructions. Briefly, 2 × 10^7^ PBMCs in 1 mL of DPBS were stained with 1 µM of CFSE in a staining buffer for 5 min at room temperature, protected from light. RPMI-1640 Medium (cat. no. R8758, Sigma-Aldrich Arklow, Ireland) containing 10% of FBS and 1% of P/S was added and incubated for 5 min at room temperature to saturate the remaining CFSE. The stained PBMCs were centrifuged at 300× *g* for 5 min and resuspended in a 1:1 mixture of RPMI-1640 medium and MSC-CM. Phytohemagglutinin (PHA) (10 µg/mL, cat. no. L8754, Sigma-Aldrich Arklow, Ireland) and recombinant interleukin-2 (cat. no. 11340023, 100 IU/mL, Immuno Tools GmbH, Friesoythe, Germany) were used to stimulate PBMC proliferation. The PBMCs were seeded at a density of 1 × 10^5^ in 100 µL mixture media per well in 96-well plates. After 96 h of culture, the PBMCs were centrifuged at 300× *g* for 5 min and resuspended in FACS buffer (DPBS with 5% FBS), and the supernatant was collected for ELISA measurement of tumour necrosis factor α (TNF-α, cat. no. DY210 R&D Systems™, Abingdon, UK), interferon γ (IFN-γ, cat. no. DY285b R&D Systems™, Abingdon, UK), and interleukin 10 (IL-10, cat. no. DY217B R&D Systems™, Abingdon, UK). The PBMCs were stained with CD3-APC-Vio770, CD4-APC, CD8-PE-Vio770, and 7-AAD (cat no. 130-113-136, 130-113-222, 130-110-680, and 130-111-568, respectively, Miltenyi Biotec, Bisley, UK) as per manufacturer’s instructions. Quantitative flow cytometry was performed using Cytek Northern Lights 3000. Data were analysed by using Flowjo software v10.6.0 (Becton Dickinson & Company, Franklin Lakes, NJ, USA).

### 2.6. Statistical Analyses

Rats were assigned randomly to sham or *E. coli* groups. Data are presented as mean ± SD. Shapiro-Wilk test was used to test for normal distribution, and data that did not pass Shapiro–Wilk test were analysed with the Mann–Whitney test. Lung injury assessment and ex vivo analyses were compared using unpaired two-tailed *t*-test or two-tailed Mann–Whitney test between two groups. BM-MSC apoptosis and activation were presented as mean percentage ± SD. Experimental groups for different concentrations of lung homogenate protein were grouped as sham and *E. coli*, and a two-way analysis of variance (ANOVA) with Šídák’s multiple comparisons test was performed to test the difference among groups. Significant differences were considered to be *p* < 0.05. A *p*-value > but close to 0.05 was also labelled in the figures. Statistical analysis was performed using GraphPad Software v9.0.0 (GraphPad Software Ltd., San Diego, CA, USA).

## 3. Results

### 3.1. E. coli Induced Injuries and Inflammation in Lungs

To identify lung injuries in the *E. coli* group, arterial oxygenation and lung compliance were measured. The mean arterial oxygenation in the *E. coli* group (76.73 ± 14.73 mmHg) was significantly lower than the mean of the sham group (127.73 ± 10.56 mmHg) (Figure 1a), and the lung compliance was lower than the sham group (0.63 ± 0.14 vs. 0.87 ± 0.081 mL/mmHg, respectively) (Figure 1b). BAL protein concentration was significantly higher in the *E. coli* group (881.17 ± 367.91 vs. 324.46 ± 172.13 µg/mL) (Figure 1c), while the protein concentration in pooled lung homogenate did not show a significant difference between groups (7464.74 ± 2846.81 vs. 5754.46 ± 2486.71 µg/mL) (Figure 1d). In BAL fluid, CINC-1, MMP-9, and IL-6 were measured, all of which were higher in the *E. coli* group compared to the sham group (Figure 1e–g). Overall, the results indicated that there is a pro-inflammatory lung microenvironment in *E. coli*-instilled animals.

### 3.2. BM-MSCs Undergo Apoptosis after Being Exposed to Lung Homogenate

Annexin V is an intracellular protein that binds to phosphatidylserine (PS) which can be translocated to the cell’s external leaflet during early apoptosis. Sytox Green dye can stain late apoptotic and necrotic cells with compromised plasma membranes. Thus, early apoptotic cells are defined as Annexin V-APC (+) and Sytox Green (-), late apoptotic cells as Annexin V-APC (+) and Sytox Green (+), and necrotic cells as Annexin V-APC (-) and Sytox Green (+). The gating strategies are shown in Figure 2a. The lung homogenate containing different concentrations of proteins (50 µg/mL, 100 µg/mL, 200 µg/mL) induced early apoptosis of BM-MSCs after 24 h of exposure compared to controls (Figure 2b). Lung homogenate from the sham group induced more apoptosis than *E. coli* groups (Figure 2b,c), but there was no dose-dependent effect. Neither lung homogenate in the sham nor *E. coli* groups induced significant necrosis of BM-MSCs within 24 h (Figure 2d).

### 3.3. The Expression of Immunomodulatory Markers on BM-MSCs Increased in Lung Homogenate Groups

CD54, also known as intercellular adhesion molecule 1 (ICAM-1), presented a higher expression level on MSCs exposed to lung homogenate from *E. coli* animals compared to control and sham groups (Figure 3a,b). CD119, also known as interferon gamma receptor 1 (IFNGR 1), presented a significantly higher median fluorescence intensity (MFI) in the *E. coli*-introduced lung homogenate stimulated group than in the control and sham groups (Figure 3c,d). Additionally, *E. coli*-infected lung homogenate increased the expression of CD200 compared to the control and sham groups (Figure 3e,f). CD120b (tumour necrosis factor receptor 2, TNFR2) and CD274 (programmed death-ligand 1, PD-L1) were both negative in all groups (Figure 3g,h). No dose-dependent effect was detected.

### 3.4. Lung Homogenate Alters the BM-MSC Secretome Profile

To profile the secretome of BM-MSCs after being exposed to lung homogenate, both pro-inflammatory cytokines, including IL-8, MCP-1, and IL-6, and anti-inflammatory cytokines, including TNFR1, TGF-β1, and VEGF-A, were measured in all groups of MSC-CM. Naïve BM-MSCs in the control group secreted very low levels of IL-8, IL-6, and MCP-1. However, the lung homogenate significantly increased the release of these cytokines in MSC-CM with a dose-dependent effect, in particular the lung homogenate from the *E. coli* group (Figure 4a–c). On the other hand, the naïve BM-MSCs in the control group secreted low levels of VEGF-A, TNFR1, and TGF-β1. The lung homogenate significantly increased the secretion of VEGF-A in MSC-CM, especially in *E. coli* groups (Figure 4d). Although lung homogenate increased the level of TNFR1 and TGF-β1, there was no significance between the control, sham, and *E. coli* groups (Figure 4e,f).

### 3.5. Lung Homogenate Maintained BM-MSCs Anti-Microbial Properties

The reduction of the OD value reflects the inhibition by MSC-CM of bacterial proliferation. Compared to MEM-α basal media, the MSC-CM significantly inhibited the proliferation of *E. coli*, *K. pneumoniae*, and *S. aureus*. BM-MSCs exposed to the lung homogenate from the sham or *E. coli* groups maintained the anti-microbial properties, but did not show enhancement compared to the control (Figure 5a–c).

### 3.6. Lung Microenvironment Conditioned MSC-CM Modulates T-Cell Proliferation In Vitro

The proliferation index and expansion index were analysed using the proliferation model in Flowjo v10.9.0. CD3+CD4+ T-cells proliferated around 5–6 generations within 96 h; the model is shown in Figure 6a. Though no significant difference was detected in the proliferation index of CD3+CD4+ T-cells among all groups, there was a significant difference in the CD3+CD4+ T-cells expansion index (Figure 6b,c). Compared to baseline (PBMC cultured in a 1:1 mixture of RPMI 1640 complete media and MEM-α basal media), MSC-CM from all groups significantly inhibited CD3+CD4+ T-cell expansion, in particular MSC-CM from lung homogenate groups. For CD3+CD8+ T-cells, MSC-CM decreased both the proliferation index and expansion index significantly in 50 µg/mL lung homogenate groups (Figure 6d–f). No significant differences were observed between the sham and *E. coli* groups.

MSC-CM from lung homogenate groups significantly decreased the levels of TNF-α, IFN-γ, and IL-10 in the PBMC culture media (Figure 6g–i), which indicates that lung homogenate can improve the MSC’s anti-inflammatory potential and is not dependent on IL-10 pathways. However, there was no significant difference between the *E. coli* and sham groups.

## 4. Discussion

The precise response of MSCs to the influence of different lung microenvironments is gradually being elucidated. Here, we found profound changes in BM-MSC biology and activity when exposed to lung homogenate, particularly *E. coli*-induced inflammatory lungs. Previous studies showed that MSCs quickly die after being trapped in lungs and are phagocytosed by monocytic cells within 24 h after infusion [23,24]. Our data shows that lung homogenate from sham animals induced more apoptosis of BM-MSCs compared to *E. coli*-infected animals within 24 h. This indicates that MSCs can retain longer in vivo under inflammatory conditions, which is consistent with previous findings [25].

Besides the apoptosis, different surface markers were also changed on MSCs exposed to lung homogenate. CD54, which plays an important role in facilitating leukocyte endothelial transmigration, can mediate MSC interaction with macrophages and improve MSC immunosuppressive capacity [26]. The surface levels of CD54 are significantly increased in MSCs exposed to lung homogenate, especially in *E. coli*-induced inflammatory lung homogenate groups. The level of CD119 (IFN-γR1), which is essential for initial IFN-γ signal transduction, is also significantly increased in inflammatory lung-primed MSCs. Similarly to CD54, CD200 on MSCs can be upregulated and interact with CD200R expressed on macrophages, modulating macrophages towards an anti-inflammatory phenotype [27]. The expression pattern of CD200 was similar to CD54 in MSCs exposed to lung homogenate. These upregulations of CD54, CD119, and CD200 indicate that BM-MSCs adapt to be more immunomodulatory in lung microenvironments, particularly in pro-inflammatory scenarios. However, the negative expression of CD120b (TNFR2) indicates that BM-MSCs may not act on the lung microenvironment in a TNF-α/TNFR2-dependent way. Additionally, the negative expression of PD-L1 suggests that BM-MSCs could not modulate activated leukocyte programmed death after being exposed to lung homogenate.

Naïve human BM-MSCs express very low levels of IL-8, IL-6 and MCP-1, while BM-MSCs exposed to lung homogenate from *E. coli*-infected animals significantly increased the level of these cytokines. IL-8 is a pro-inflammatory cytokine, reducing the therapeutic effects of MSCs in acute-on-chronic liver failure (ACLF) [28]. Thus, the increase of IL-8 from pro-inflammatory lung homogenate-primed MSCs suggests a defect in current MSC therapy and a strategy to enhance MSC efficacy through IL-8 blocking. IL-6 plays a dual role in the MSC’s immunosuppressive function: while MSC-derived IL-6 can delay apoptosis of lymphocytes [29], which can promote inflammation, IL-6 is also essential for the MSC-mediated differentiation of monocyte-derived cells towards an IL-10-producing phenotype [30]. MSC-derived MCP-1 has been reported to promote T-cell proliferation and differentiation by inhibiting PD-L1 production [31]. Lung homogenate from *E. coli*-infected animals significantly induced the release of MCP-1 from MSCs, indicating that the MSC’s immunosuppressive effect is partly inhibited in the lung microenvironment. On the other hand, MSCs exposed to lung homogenate expressed higher levels of VEGF-A, which plays a vital role in the repair of the endothelial barrier. The MSC secretome significantly changed after being exposed to lung homogenate. By profiling these changes, which reflect MSC behaviour under normal and inflammatory conditions, a more comprehensive understanding of MSC function can be obtained.

Our data support the theory that MSC-CM can inhibit the proliferation of *E. coli*, *K. pneumoniae*, and *S. aureus* in vitro. The exposure to lung homogenate maintained these effects, which indicates that soluble factors derived from MSCs exert anti-microbial effects under both healthy and bacteria-infected conditions.

MSCs can inhibit T-cell proliferation through cell–cell contact and soluble mediators, which is dependent on cross-talk between T-cells and MSCs [32]. Here, we found that without the crosstalk, MSC-CM did not significantly decrease the proliferation index of the CD3+CD4+ T-cells, but significantly decreased the expansion index. The proliferation index is the total division number of cells that have commenced division, excluding undivided cells, while the expansion index indicates the total divided cell number compared to cells at the start of culture. Thus, the results indicate that MSC-CM could not inhibit the proliferation of the activated CD3+CD4+ T-cells but suppress the CD3+CD4+ T-cell switch to a proliferation state. For CD3+CD8+ T-cells, only MSC-CM from the 50 µg/mL group decreased both the proliferation and expansion index significantly, showing a dose effect that may be related to the higher IL-6 and MCP-1 in MSC-CM from 100 µg/mL and 200 µg/mL groups. Additionally, MSC-CM inhibited the release of TNF-α and IFN-γ, which play important roles in T-cell proliferation and activation [33,34]. The decrease in IL-10 suggests that MSC-CM inhibition of T-cell proliferation was not IL-10 dependent.

In summary, exposure to either a healthy or an inflammatory lung microenvironment induces BM-MSCs apoptosis and changes in their phenotype and secretome. These changes not only imply differential BM-MSC reaction to healthy and inflammatory environments, but also indicate further strategies to improve the therapeutic effects of MSCs targeting different mechanisms. After exposure to the lung microenvironment, BM-MSCs undergo apoptosis and express more CD54, CD119, and CD200 to engage in immunomodulation. Though the soluble mediators secreted by BM-MSCs exposed to lung microenvironment are not detrimental to their anti-microbial properties, both pro-inflammatory and anti-inflammatory mediators increased in the secretome of BM-MSCs, which may play a dual role in inhibiting T-cell proliferation. These findings provide new insights into MSC-based cell therapy for ARDS and give a broader understanding of the MSC’s complex reactions to different lung microenvironments, suggesting further strategies to improve MSC therapeutic effects.

There are limitations to this study. Like serum or BAL used in previous studies, lung homogenate can mimic the molecular microenvironment in healthy and ARDS lungs but cannot imitate the cell–cell crosstalk between infused MSCs and the lung microenvironment. After being infused, MSC kinetics cannot be detected through this study either. Previous studies have demonstrated controversial results regarding whether soluble factors from MSCs are efficient at inhibiting T-cell proliferation [35,36,37], and even though there were significant differences in T-cell proliferation and expansion indexes in this study, more PBMCs from different donors will be needed to validate these results. To develop advanced MSC-based therapies against ARDS, well-established animal models will be necessary for exploring the MSC’s fate and behaviours in vivo. Moreover, different sources of MSCs and more strategies to modulate MSCs need to be investigated to improve clinical outcomes.

## 5. Conclusions

The lung microenvironment influences the activity of a live cell therapy such as the MSC, and this interaction is different depending on the disease state of the host. These findings could inform future enhanced MSC therapeutic approaches for ARDS.

## Figures and Tables

**Figure 1 cells-13-01581-f001:**
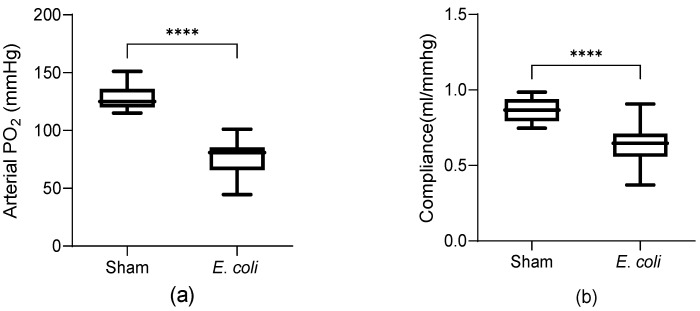
Assessment of lung injury and inflammation in the sham (*n* = 11) and *E. coli* groups (*n* = 13). (**a**) Arterial oxygenation with 21% inspired oxygen was significantly lower in the *E. coli* group. (**b**) Lung compliance was significantly lower in the *E. coli* group compared to sham. (**c**) Protein concentration in BAL fluid was significantly higher in the *E. coli* group. (**d**) Protein concentration in pooled lung homogenate from the sham and *E. coli* groups showed no significant difference. (**e**) The level of CINC-1 in BAL fluid was significantly higher in the *E. coli* group (50.85, 27.66–72.69 vs. 190.0, 144.5–240.7 pg/mL). (**f**) The level of MMP-9 in BAL fluid was significantly higher in the *E. coli* group (91.05, 3.774–384.5 vs. 9969, 8382–17908 pg/mL). (**g**) The level of IL-6 in BAL fluid was significantly higher in the *E. coli* group (108.49 ± 102.72 vs. 549.60 ± 240.53 pg/mL). Results are shown as the minimum, first quartile, median, third quartile, and maximum, and analysed by two-tailed *t*-tests: ***: *p* < 0.001; ****: *p* < 0.0001, or two-tailed Mann–Whitney test: ###: *p* < 0.001; ####: *p* < 0.0001. “ns” denotes non-significant differences.

**Figure 2 cells-13-01581-f002:**
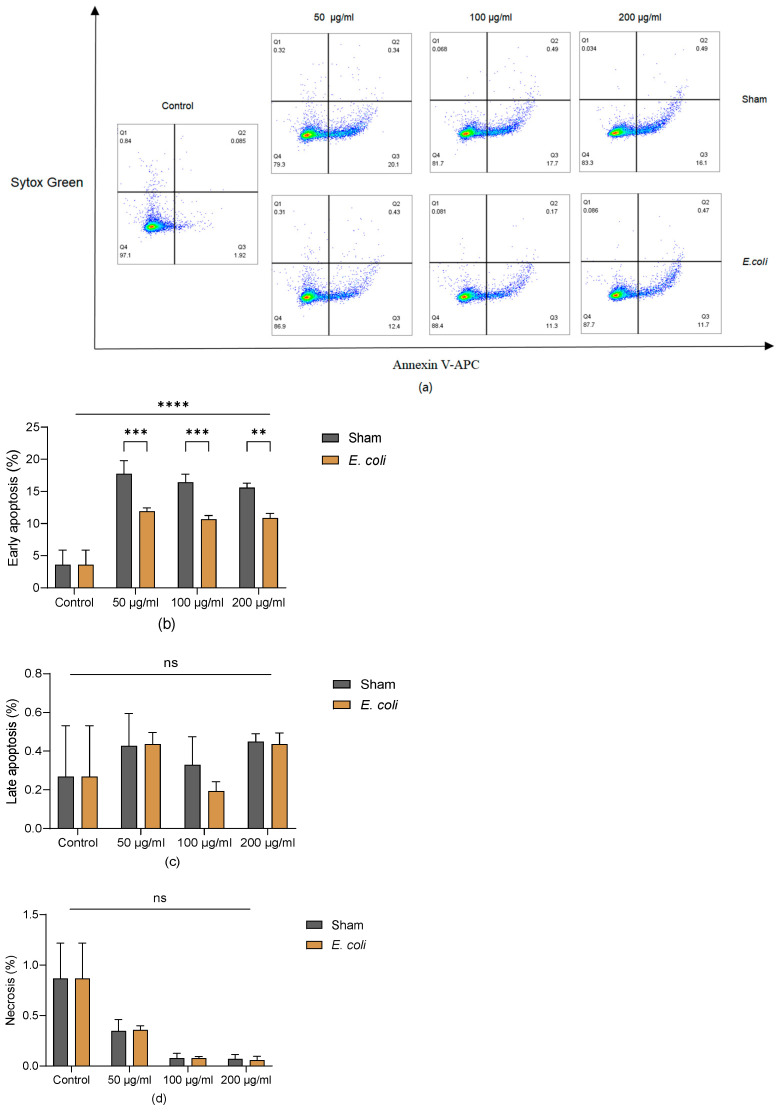
Apoptosis of BM-MSC after exposure to pooled lung homogenate. (**a**) Gating strategies with Annexin V-APC and Sytox Green. (**b**) Compared to the control (BM-MSCs cultured without lung homogenate), lung homogenate significantly induced apoptosis of BM-MSCs, especially in sham groups. In (**c**,**d**), the level of late apoptosis and necrosis of BM-MSC after exposure to lung homogenate in all groups was lower than 1%. Representative results of three independent experiments are shown as mean ± SD and analysed by two-way ANOVA with Šídák’s multiple comparisons test. **: *p* < 0.01; ***: *p* < 0.001; ****: *p* < 0.0001; ns: no significance.

**Figure 3 cells-13-01581-f003:**
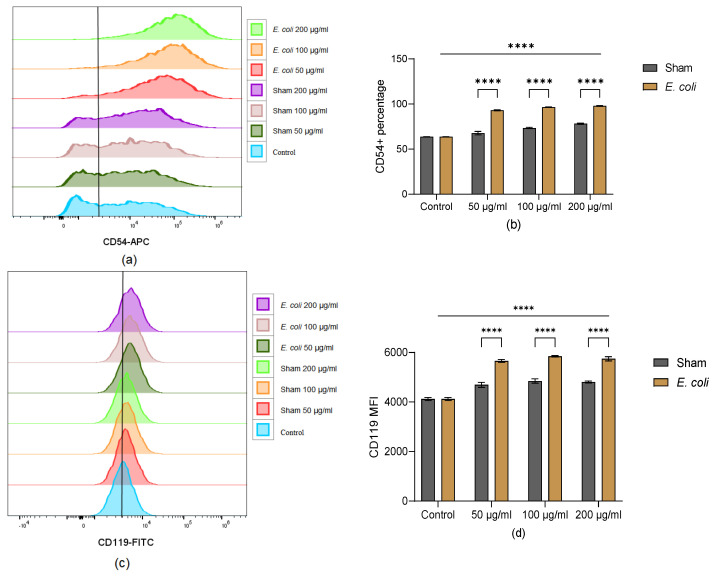
Activation of BM-MSC after exposure to pooled lung homogenate. (**a**,**b**) Expression level of CD54 (positive percentage) in viable BM-MSCs: CD54 positive percentage was significantly higher in lung homogenate groups, especially *E. coli* groups. (**c**,**d**) Expression level of CD119 (MFI) of viable BM-MSCs: CD119 MFI was significantly higher in *E. coli* groups than in the control and sham groups. (**e**,**f**) Expression level of CD200 (positive percentage) of viable BM-MSCs: CD200 positive percentage was significantly higher in lung homogenate groups, particularly in *E. coli* groups. (**g**,**h**) Negative expression of CD120b and CD274 on BM-MSCs in each group (**g**,**h**). Representative results of three independent experiments are shown as mean ± SD and analysed by two-way ANOVA with Šídák’s multiple comparisons test. ***: *p* < 0.001; ****: *p* < 0.0001.

**Figure 4 cells-13-01581-f004:**
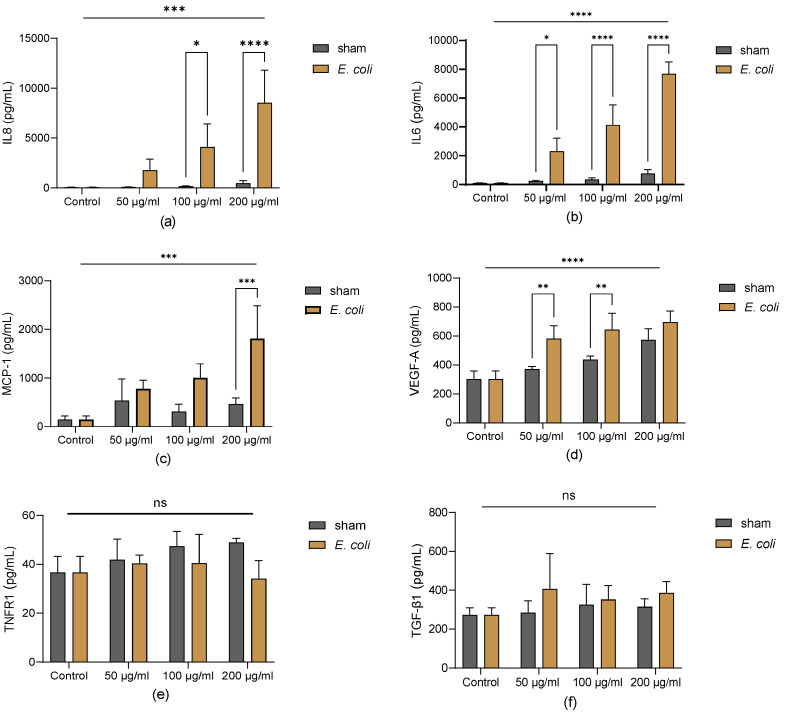
Cytokine measurement in MSC-CM exposed to pooled lung homogenate. (**a**) Lung homogenate exposure significantly increased IL-8 concentration in MSC-CM; the level of IL-8 was significantly higher in *E. coli* groups at 100 and 200 µg/mL than in sham groups at the same concentrations. (**b**) The level of IL-6 in MSC-CM was significantly higher in the *E. coli* groups than in the control and sham groups at all concentrations. (**c**) Lung homogenate exposure significantly increased the release of MCP-1 in MSC-CM; the level of MCP-1 was significantly higher in *E. coli* groups at 200 µg/mL than in sham groups at the same concentration. (**d**) Lung homogenate exposure significantly increased the level of VEGF-A in MSC-CM; the level of VEGF-A was significantly higher in *E. coli* groups at 50 and 100 µg/mL than in sham groups at the same concentrations. (**e**,**f**) The level of TNFR1 and TGF-β1 in MSC-CM: No significant difference was detected between any groups. Representative results of three independent experiments are shown as mean ± SD and analysed by two-way ANOVA with Šídák’s multiple comparisons test. *: *p* < 0.05; **: *p* < 0.01; ***: *p* < 0.001; ****: *p* < 0.0001; ns: no significance.

**Figure 5 cells-13-01581-f005:**
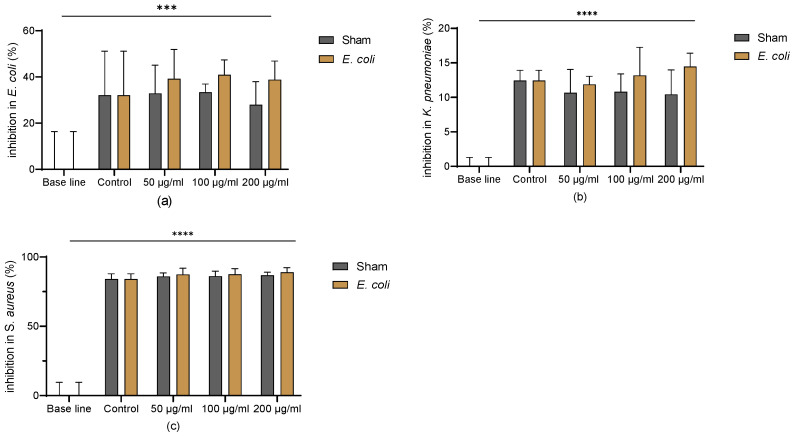
Anti-microbial properties of CM from MSC exposed to pooled lung homogenate. (**a**) MSC-CM in all groups significantly inhibited the proliferation of *E. coli*. (**b**) MSC-CM in all groups significantly inhibited the proliferation of *K. pneumoniae*. (**c**) MSC-CM in all groups significantly inhibited the proliferation of *S. aureus*. Representative results of three independent experiments are shown as mean ± SD and analysed by two-way ANOVA with Šídák’s multiple comparisons test. ***: *p* < 0.001; ****: *p* < 0.0001.

**Figure 6 cells-13-01581-f006:**
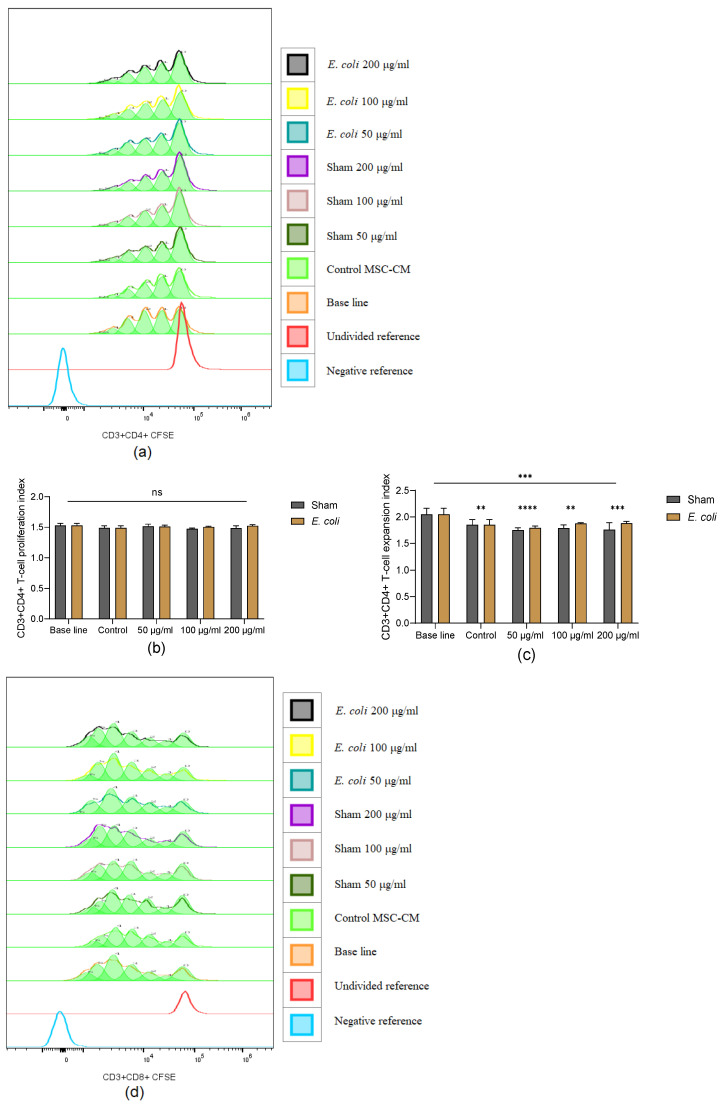
T-cell proliferation with MSC-CM conditioned with pooled lung homogenate. (**a**) The proliferation model of CD3+CD4+ T-cells with Flowjo: CD3+CD4+ T-cells went through 5–6 generations. (**b**) The proliferation index of CD3+CD4+ T-cells: no significant differences were detected. (**c**) The expansion index of CD3+CD4+ T-cells: MSC-CM significantly inhibited CD3+CD4+ T-cells expansion, particularly in lung homogenate groups. (**d**) The proliferation model of CD3+CD8+ T-cells: CD3+CD8+ T-cells proliferated 5–6 generations. (**e**,**f**) The proliferation index and expansion index of CD3+CD8+ T-cells: MSC-CM inhibited CD3+CD8+ T-cells proliferation and expansion significantly in 50 µg/mL lung homogenate groups. MSC-CM from all lung homogenate groups significantly decreased the level of TNF-α (**g**), IFN-γ (**h**), and IL-10 (**i**) in PBMC culture media. Representative results of three independent experiments are shown as mean ± SD and analysed by two-way ANOVA with Šídák’s multiple comparisons test. *: *p* < 0.001; **: *p* < 0.01; ***: *p* < 0.001; ****: *p* < 0.0001; ns: no significance.

## Data Availability

All raw data is available upon request.

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
