# Peer review of "The Effects of the Pneumonia Lung Microenvironment on MSC Function"

_cells, 2024, doi:10.3390/cells13181581_

Round 1

Reviewer 1 Report (Previous Reviewer 1)

Comments and Suggestions for Authors

1) In the introductions section it will better to star with other  experimental studies and pre-clinical studies, and then, mention clinical studies (about security and efficacy).

2) Justify why the sham group had 11 animals and the experimental 13. Why both were not 13?

3) Material and methods, are nice written.

4) Which statistical test was performed to determine data distribution?

5) In figure 1, boxes show clearly a non-parametric distribution. Therefore, authors should reporte median value and percentile 25-75. Also with this distribution T-tests is NOT adequate, authors should performed non-parametric test like Mann-Whitney U test. 

6) If in figure 1, data was not parametric, it's difficult that data from figures 2, 3, 4, 5 and 6will be parametric. Please provide information about data distribution of figure 3. Particularly in figure 4 and 6.  

7) We have serous doubts about the data distribution and the correct statistical analysis. 

Author Response

Reviewer 1: Comments and Suggestions for Authors

1) In the introductions section it will better to star with other experimental studies and pre-clinical studies, and then, mention clinical studies (about security and efficacy).

> Thanks for your suggestion. In Introduction paragraph 1, we described the mechanisms by which MSCs attenuate lung injury in animal models and mentioned the main issues with MSCs therapy in clinical trials to point out the importance of investigating how the microenvironment affects MSC function. In paragraph 2, we described what previous studies have found about the interaction between host microenvironment and MSC function. To get a better flow, we deleted “in preclinical studies” in the first sentence because it’s not well connected with the studies we described in paragraph 2.

2) Justify why the sham group had 11 animals and the experimental 13. Why both were not 13?

> At the commencement of the study 13 animals were assigned to both groups, but two animals from the sham group died before assessment procedures were performed. These were due to intratracheal instillation and intracarotid artery cannulation problems. As there were no further animals available without a new experimental license application under our national regulations, we proceeded with statistical calculations based on these results. We regret this loss of possible additional robustness of statistical confidence.

3) Material and methods, are nice written.

> Thanks for your comments.

4) Which statistical test was performed to determine data distribution?

> We assumed the data is normal distributed as variables in natural sciences are normally or approximately normally distributed, like height, weight or soluble factors in serum etc., but as you recommend, we did normality test for our data with Shapiro-Wilk test which is one of the most commonly used normality tests and we found that most of them are normally distributed.

5) In figure 1, boxes show clearly a non-parametric distribution. Therefore, authors should reporte median value and percentile 25-75. Also with this distribution T-tests is NOT adequate, authors should performed non-parametric test like Mann-Whitney U test.

> Even though not all the median lines are exactly in the middle of the boxes, it does not mean they are not normally distributed. We confirmed normality for our data with the Shapiro-Wilk test and found that except for BAL CINC1 and MMP9, all other parameters were normally distributed. According to the results from this normality test, we present the median value and percentile 25-75 of CINC1 and MMP9 in the caption and used Mann-Whitney test to compare the two groups.

6) If in figure 1, data was not parametric, it's difficult that data from figures 2, 3, 4, 5 and 6 will be parametric. Please provide information about data distribution of figure 3. Particularly in figure 4 and 6.

> We performed the Shapiro-Wilk test for all our data; the results are as follows.

Figure 2: (a) all groups passed normality test except for sham 50 µg/mL; (b) all groups passed normality test; (c) all groups passed normality test.

Figure 3: (b) all groups passed normality test except for sham 50 µg/mL; (d) all groups passed normality test; (f) all groups passed normality test.

Figure 4: all the cytokines in all groups passed normality test except for IL-8 in group E. coli 200 µg/mL.

Figure 5: all groups passed normality test.

Figure 6: all groups passed normality test.

We found that in the groups that did not pass normality test there was always an outlier, which can explain the phenomenon. Besides, if there is same data in one group, the test loses its power. Since we have three replicates which meets the lowest number to do the Shapiro-Wilk test, it is reasonable to assume that all our data is normally distributed, and Two-ways ANOVA is suitable to analyse these data.

7) We have serous doubts about the data distribution and the correct statistical analysis.

> Normality tests were performed for all our data and we believe the results support the statistical analysis.  All Normal QQ plots from Shapiro-Wilk test are attached in Supplementary Data Normality test.doc. for further inspection.

Reviewer 2 Report (New Reviewer)

Comments and Suggestions for Authors

Thank you for the opportunity to review this interesting and well written original study with appropriate methodology, following new insights into MSC-based cell therapy for ARDS and give a broader understanding of MSCs complex actions to different lung microenvironment. Authors in this original research article assess the effects of a new aspect of normal and ARDS lung microenvironments on an administered MSC therapeutic.

Some minor comments follow: Further enrichment of the introduction is suggested. Authors could further mention the key role of mesenchymal stromal cell interaction with macrophages in promoting repair of lung injury. MSCs are now known to enhance the repair of injured tissues and are emerging as possible therapeutic agents in acute and chronic inflammatory lung diseases and in ARDS. This recently published article which summarizes current knowledge for the potential interaction of MSCs in lung repair in inflammatory lung diseases could be provided in the reference list i.e. " Jerkic, M.; Szaszi, K.; Laffey, J.G.; Rotstein, O.; Zhang, H. Key Role of Mesenchymal Stromal Cell Interaction with Macrophages in Promoting Repair of Lung Injury. Int. J. Mol. Sci. 2023, 24, 3376."

Author Response

Reviewer 2: Comments and Suggestions for Authors

Thank you for the opportunity to review this interesting and well written original study with appropriate methodology, following new insights into MSC-based cell therapy for ARDS and give a broader understanding of MSCs complex actions to different lung microenvironment. Authors in this original research article assess the effects of a new aspect of normal and ARDS lung microenvironments on an administered MSC therapeutic.

Some minor comments follow: Further enrichment of the introduction is suggested. Authors could further mention the key role of mesenchymal stromal cell interaction with macrophages in promoting repair of lung injury. MSCs are now known to enhance the repair of injured tissues and are emerging as possible therapeutic agents in acute and chronic inflammatory lung diseases and in ARDS. This recently published article which summarizes current knowledge for the potential interaction of MSCs in lung repair in inflammatory lung diseases could be provided in the reference list i.e. " Jerkic, M.; Szaszi, K.; Laffey, J.G.; Rotstein, O.; Zhang, H. Key Role of Mesenchymal Stromal Cell Interaction with Macrophages in Promoting Repair of Lung Injury. Int. J. Mol. Sci. 2023, 24, 3376."

> Thanks for your comments. We have extended the description regarding how MSCs exert their anti-inflammatory effects through MSC-macrophage crosstalk and cited the related literature.

Round 2

Reviewer 1 Report (Previous Reviewer 1)

Comments and Suggestions for Authors

Data from figure 1 shows non parametric distribution, but the authors performed a T test. Is not correct, Also reportes Mann-Whintey U test, this kind of data only should be tested by a Mann-Whintey no T-test. 

For figure 2, 3 and 4, one group did not have normal distribution, therefore the tests should be no parametric. Also the column graphic is not the correct one. 

Author Response

Dear Reviewer,

  1. Data from figure 1 shows non parametric distribution, but the authors performed a T test. Is not correct, Also reportes Mann-Whintey U test, this kind of data only should be tested by a Mann-Whintey no T-test.

> We performed Shapiro-Wilk test for normal distribution and only data of BAL CINC1 and MMP9 did not pass this test. Therefore, we did Mann-Whitney test for data of CINC2 and MMP9, “#” was used to show statistical differences on these two graphs to distinguish from t test.

  1. For figure 2, 3 and 4, one group did not have normal distribution, therefore the tests should be no parametric. Also the column graphic is not the correct one.

> The QQ plots from Shapiro-Wilk test showed that except for outliers, all the data is normal distribution, which supports our data analyses methods.

Regards

This manuscript is a resubmission of an earlier submission. The following is a list of the peer review reports and author responses from that submission.

Round 1

Reviewer 1 Report

Comments and Suggestions for Authors

1) Introduction is vague about the physiopathology of ARDS and the possible mechanism or effect over MSC. 

2) Material and methods are nice described. 

3) The n (sample size) are 3 animals? If that is true were are the replicate and reproducibility of their experiments? Line 201 and figure 1D, why just a n of 3? If is this true, eliminate figure 1d. 

4) The same problem with figure 2 to 67 the authors mention an n=3. This is not enough. Where is the reproducibility of make at least twice de experiment?

At least the author should make an experiment with three animals, twice. If the variation between first and second experiment is higher than 10-20%, should performed a third experiment. 

Reviewer 2 Report

Comments and Suggestions for Authors

The paper is overall well written and suitable to the journal. The only concern regards the lack of data, in animal model , of pharmacy-kinetics. I would suggest to deeply investigate this point that should be relevant for the next future improvement of advanced cell therapies against lung damages.

Comments on the Quality of English Language

The quality of English is good, only minor revision required.